

# Clonidine or remifentanil for adequate surgical conditions in patients undergoing endoscopic sinus surgery: a randomized study

Laurent Bairy[1], Marie Vanderstichelen[2], Jacques Jamart[3] and Edith Collard[1]

[1] Department of Anesthesiology, CHU UcL Namur, Yvoir, Belgium
[2] Department of Anesthesiology, Cliniques Universitaires Saint-Luc, Brussels, Belgium
[3] Department of Biostatistics, CHU Ucl Namur, Yvoir, Belgium

## ABSTRACT

**Background**. Deliberate hypotension is one way to achieve a bloodless surgical field in endoscopic sinus surgery (ESS). We compared two anaesthesia regimens to induce deliberate hypotension and attempted to determine the most efficient one.

**Methods**. Fifty-nine patients undergoing ESS were minimized into two groups. In the CLO group, patients received I.V. sufentanil 0.15 µg/kg together with I.V. clonidine 2–3 µg/kg. In the REMI group, patients received remifentanil at a rate of up to 1 µg/kg/min. Fromme scores were collected 15 min after the incision and at the end of the procedure. Mean arterial pressure readings (MAP), heart rate readings, time to eyes opening, time to extubation, pain scores, analgesic requirements, and oxygen needs were collected and compared.

**Results**. There were no significant differences in Fromme scores between the two groups. The averaged MAP from 15 min to the end of the procedure was significantly lower in the REMI group; these patients also received more ephedrine. Significantly fewer patients in the CLO group needed oxygen therapy to keep their Pulse Oximeter Oxygen Saturation within 3% of their preoperative values. Patients in this group also needed less piritramide in the recovery room, and their pain scores were lower at discharge from the recovery room.

**Discussion**. Although both anaesthesia regimens offered a similar quality of surgical field, this study suggests that clonidine had a better average safety profile. Furthermore, patients who received this regimen required fewer painkillers immediately after surgery.

## INTRODUCTION

Over the past decades, endoscopic sinus surgery (ESS) has developed into one of the most frequently used surgical treatments for chronic inflammatory or infectious sinus diseases. This procedure requires a bloodless surgical field to help the surgeon visualize anatomical structures to avoid injuries and rare, but potentially catastrophic, complications (*Baker & Baker, 2010*).

Corresponding author
Laurent Bairy,
laurent.bairy@uclouvain.be

To achieve a bloodless surgical field, deliberate hypotension is often used among other interventions, such as preoperative oral corticosteroid treatment prescribed in case of severe disease, moderate hypocapnia, reverse Trendelenburg position, infiltration with a local anesthetic solution containing a vasopressor and/or topical anesthesia of the surgical site.

Deliberate hypotension has been used in numerous procedures to decrease blood loss or increase surgical visibility. There are multiple drugs and techniques that can be used to induce such deliberate hypotension, from anesthesia drugs that have hypotension as side effect (volatile anesthetics, remifentanil) to specific hypotensive agents such as nitroprusside, nicardipine, adenosine, nitroglycerine or esmolol. Some antihypertensive drugs are also used such as ACE inhibitors or clonidine (*Degoute, 2007*).

Clonidine is an antihypertensive drug introduced in the 1970s. It is used increasingly in the perioperative setting due to its desirable properties such as morphinic and hypnotic sparing effect, cardioprotection, anxiolysis, reduced shivering, reduced perioperative stress response (*Wallace, 2006*). Often, it lowers arterial pressure down to values compatible with the definition of controlled hypotension when used in conjunction with volatile anesthetic or propofol. It has already been used in multiple studies related to deliberate hypotension. Remifenanil is an ultrashort acting μ agonist. It produces consistent hypotension as a side effect. These two drugs are the most used in our institution to induce deliberate hypotension in case of ESS. The aim of this study is to compare two anesthesia regimens that include these drugs and determine which regimen gives the best operating conditions to the surgeon. Other data collected and compared were emergence duration, pain intensity, hemodynamic and oxygenation ($SpO_2$) parameters.

In this study, deliberate hypotension is defined by a target mean arterial pressure (MAP) between 55 and 65 mmHg.

## METHODS

### Patients

All patients with chronic sinus disease, older than 18 years, categorized ASA physical status I to III, primary or secondary surgery were assessed for eligibility. Exclusion criteria included cardiac disorders other than supraventricular tachycardia, cerebrovascular disorders, renal or hepatic disorders, non-treated arterial hypertension, beta-blocking agent therapy, platelet-inhibiting agent or anticoagulant therapy, coagulopathy, and pregnancy. Fifty-nine patients were enrolled in the study between November 2014 and March 2015 at the CHU UcL Namur, a tertiary care center in Belgium. Patients were prospectively randomized in the CLO group or the REMI group using minimization (*Saghaei & Saghaei, 2011*) the day before or the day of the surgery by the main investigator (Fig. 1). The factors used for minimization included age, sex, Mackay-Lund score that assesses the extent of the sinus disease on a sinus CT scan (*Fokkens, Lund & Mullol, 2007*), preoperative oral corticoid treatment, ASA score and one of the two possible surgeons.

### Treatment

In the CLO group, anesthesia was induced with clonidine 2 μg/kg slow I.V. bolus over 10 min, sufentanil 0.15 μg/kg, propofol titrated to loss of eyelid reflex and rocuronium
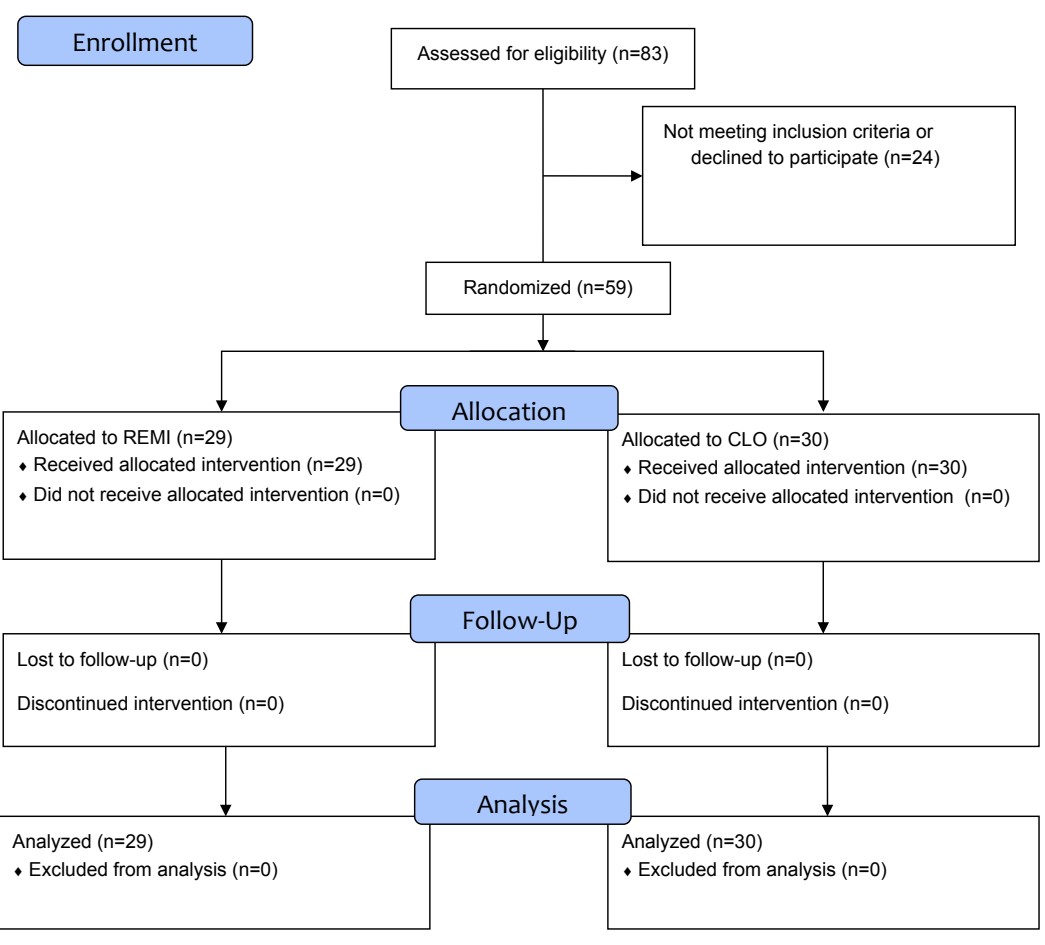

**Figure 1 CONSORT flow diagram.**

0.3 mg/kg. In the REMI group, anesthesia was induced with a remifentanil I.V. infusion of 0.25 µg/kg/min, propofol titrated to the loss of eyelid reflex, and rocuronium 0.3 mg/kg. The remifentanil infusion rate was halved after tracheal intubation. In both groups, the trachea was intubated and the anesthesia was maintained using sevoflurane 1 MAC (age corrected) in a 50% $FiO_2$ oxygen/air mixture. The gas mixture was delivered to the patient by a Dräger Zeus$^{TM}$ (Dräger Medical, Lübeck, Germany) ventilator in a circle circuit using the "auto-controlled mode". In both groups, the following settings were used for mechanical ventilation: 3 cm $H_2O$ PEEP, 6–8 ml/kg tidal volume and respiratory rate of 12/min. These parameters were subsequently adjusted to maintain $EtCO_2$ between 30 and 35 mmHg. The patients were positioned 15° reverse Trendelenburg. Non-invasive blood pressure, heart rate (HR), transcutaneous oxygen saturation ($SpO_2$) and train of four (TOF) were measured using a Philips MX800 monitor (Phillips Medical, Nijmegen, the Netherlands). These parameters, along with respiratory parameters from the ventilator, were recorded by the Exacto$^{TM}$ computerized anesthesia record system (Mexys, Mons, Belgium). Local infiltration with lidocaine 1% and epinephrine 1/200000 was performed and followed by a topical application of lidocaine and naphazoline for mucosal anesthesia

**Table 1  Fromme score.**

| |
|---|
| 5 Massive uncontrollable bleeding. |
| 4 Bleeding, heavy but controllable, that significantly interfered with dissection. |
| 3 Moderate bleeding that moderately compromised surgical dissection. |
| 2 Moderate bleeding, a nuisance but without interference with accurate dissection. |
| 1 Bleeding, so mild it was not even a surgical nuisance. |
| 0 No bleeding, virtually bloodless field. |

and vasoconstriction. If the surgeon complained about the bleeding or if the Fromme score (*Fromme et al., 1986*) (Table 1) was 3, 4 or 5 and the MAP was higher than 65 mmHg, the patients in the CLO group received an additional 1 μg/kg slow I.V. clonidine bolus (up to a maximum total dose of 3 μg/kg) or a sufentanil 5 μg I.V. bolus if the previous one was administered more than 45 min before. In the REMI group, the remifentanil infusion rate was doubled up to a maximum rate of 1 μg/kg/min. Despite these measures, if the surgeon still noted considerable bleeding, the patient was excluded from the study.

If the MAP was lower than 55 mmHg, ephedrine was titrated in 3-mg increments to achieve a MAP higher than 55 mmHg. Additionally, in the REMI group, the remifentanil infusion rate was halved. In case of failure to maintain a MAP higher to 55 mmHg using ephedrine, the patient was excluded from the study.

Sevoflurane and remifentanil, if applicable, were stopped when the surgeon began packing the nose and the patient's TOF ratio attained 0.9. Patients who had still a TOF ratio lower than 0.9 were given glycopyrrolate 0.5 mg and neostigmine 2.5 mg. Analgesia was provided by giving the patient paracetamol 1 g I.V., tramadol 100 mg I.V. and ketorolac 30 mg I.V., respecting contraindications and dosage limitations for each patient, if appropriate. All patients received 50 mg alizapride I.V. to prevent nausea and vomiting. In the recovery room, oxygen was delivered through a facemask to keep the patient's $SpO_2$ within 3% of the preoperative value.

The patient was discharged from the recovery room when his/her modified Aldrete score was ≥9 and when the patient was comfortable.

## Assessment

We decided not to use blood loss as an indicator, as it is very difficult to measure accurately due to the large amount of irrigation fluid used in this type of surgery and, more often than not, it is minimal. Blood loss was grossly estimated to ensure that the patient did not reach the transfusion threshold.

The quality of the surgical field was assessed by the surgeon (blinded), using the Fromme score 15 min after the start of the surgery (From15) to allow the MAP and HR reach an equilibrium after the local anesthetic infiltration and the first surgical stimulus. The Fromme score at the end of the surgery (FromE) gave an overall rating of the surgical field quality. The quality of the surgical field was characterized as low when the Fromme score was higher than 2 or high if the score was lower or equal to 2.

In the recovery room, pain was evaluated using the 11-point numerical rating scale (NRS-11), which is a patient self-reported pain score that ranges from 0 implying no pain

to 10 implying the worst pain imaginable. If the patient reported a NRS-11 higher than 3, piritramide was administered in 2 mg increments.

The following parameters were also recorded and compared: continuous $SpO_2$ and HR, MAP every 3 min during the surgery and every 10 min in the recovery room, time to eyes opening (time patient opened his/her eyes− time when sevoflurane ± remifentanil was/were stopped), time to extubation (time patient was extubated− time sevoflurane ± remifentanil was/were stopped), time spent in the recovery room, pain scores at the arrival and departure from the recovery room, piritramide consumption in the recovery room and need for oxygen when patient returned to the ward.

Averaged MAP readings (MAPA) and averaged HR (HRA) values from 15 min after the onset of the procedure to the end of the surgery were computed using the computerized anesthesia record system database.

## Statistics

The sample size was computed to allow detection of a difference between Fromme scores of 1 point, with a standard deviation of 1.3 estimated from a preliminary assessment of the Fromme score on 40 patients, with an $\alpha$ level of 0.05 and a power $(1-\beta)$ of 0.80. The computation used G*Power 3 software (Heinrich Heine Universität, Dusseldorf, Germany) (*Faul et al., 2007*) and led to the total sample size of 58 patients.

All the analyses were performed with the intention to treat.

Statistical analyses were carried out using SPSS$^{TM}$ software (v18.0, SPSS Inc., Chicago, IL, USA).

As we were not able to claim the normality of all numerical variables, their comparisons were performed using the Wilcoxon–Mann–Whitney test, and categorical variables were compared using Fisher's exact test. A *p*-value of 0.05 was considered significant and was not corrected for the number of tests performed.

## Ethical statement

This study was approved by the local ethics committee (ID 101/2003. Belgian unique ID B039201318956). It was prospectively registered in the Australian and New Zealand clinical trial register under ID ACTRN12614000935639. A written informed consent was obtained from all the patients.

## RESULTS

No patient was excluded from the study.

There were no significant differences between both groups in regard to body mass index (BMI), preoperative MAP, HR, $SpO_2$ values, and duration of surgery. In the CLO group, 36.7% were smokers compared to 6.9% in the REMI group ($p = 0.01$) (Table 2).

In the CLO group, 20 patients received clonidine 2 μg/kg I.V. and 9 patients received 3 μg/kg I.V. The mean ± SD sufentanil dose was 0.16 μg/kg ± 0.03 μg/kg. In the REMI group, patients received 0.34 μg/kg/min ± 0.17 μg/kg/min (mean ± SD) of remifentanil.

There were no significant differences in the From15 or FromE scores between the two groups (Table 3). The quality of the surgical field was deemed low in nine patients in

**Table 2  Patients' demographic data.**

|  | CLO group ($n = 30$) | REMI group ($n = 29$) | $p$ |
|---|---|---|---|
| Age (mean $\pm$ SD) | 46 $\pm$ 15 | 46 $\pm$ 15 | |
| Sex M/F | 13/17 | 13/16 | |
| Surgeon A/B | 10/20 | 11/18 | |
| ASA I/II/III | 6/24/0 | 3/25/1 | |
| Preop. oral corticoids | 6.7% | 13.8% | |
| Mackay-Lund score (median $\pm$ IQR) | 7.4 $\pm$ 5.1 | 7.2 $\pm$ 5.9 | |
| BMI (mean $\pm$ SD) | 25.9 $\pm$ 4.7 | 27.1 $\pm$ 5.4 | 0.39 |
| Smokers | 36.7% | 6.9% | 0.01[*] |

**Notes.**
[*]Statistically significant.

**Table 3  Outcomes expressed as median and interquartile range (IQR) (Wilcoxon–Mann–Whitney test).**

|  | CLO group | | REMI group | | $p$ |
|---|---|---|---|---|---|
|  | Median | IQR | Median | IQR | |
| From15 score | 2 | 1.75 | 2 | 1 | 0.58 |
| FromE score | 3 | 1 | 2 | 1 | 0.24 |
| Pain NRS at recovery room admission | 0 | 0 | 0 | 0 | 0.21 |
| Pain NRS at recovery room discharge | 0 | 1 | 1 | 2 | 0.04[*] |
| Preoperative SpO$_2$ (%) | 100 | 1 | 100 | 3 | 0.64 |
| Preoperative MAP (mmHg) | 87.5 | 17.5 | 90 | 19 | 0.50 |
| Preoperative HR (bpm) | 67.5 | 13 | 73 | 16 | 0.19 |
| MAP at 15 min (mmHg) | 61 | 14.75 | 56 | 10 | 0.10 |
| HR at 15 min (bpm) | 68 | 21.75 | 62 | 18 | 0.12 |
| Mean MAP from 15 min after starting to end of surgery (mmHg) | 61.5 | 8.5 | 57 | 8 | 0.04[*] |
| Mean HR from 15 min after starting to end of surgery (mmHg) | 64.5 | 18.25 | 64 | 12 | 0.35 |
| Ephedrine total dose (mg) | 0 | 11.25 | 9 | 15 | 0.01[*] |
| Length of surgery (min) | 53 | 39 | 57 | 41 | 0.63 |
| Time to eyes opening (min) | 16 | 8 | 14 | 8 | 0.43 |
| Time to extubation (min) | 10 | 7 | 13 | 6 | 0.64 |
| Recovery room stay length (min) | 74 | 24 | 76 | 33 | 0.66 |
| Piritramide dose in the recovery room (mg) | 0 | 0 | 0 | 4 | 0.02[*] |
| Mean SpO$_2$ in the recovery room (%) | 98 | 2 | 98 | 2 | 0.21 |

**Notes.**
[*]Statistically significant.

the CLO group compared to 11 patients in the REMI group at 15 min ($p = 0.59$) and 16 patients in the CLO group compared to 10 patients in the REMI group at the end of the surgery ($p = 0.19$).

The MAP and HR at 15 min after incision were similar in both groups. HRA did not differ between the two groups. The MAPA was significantly lower in the REMI group. The total amount of ephedrine administered to the patients was significantly higher in the REMI group, as 79.3% of these patients needed ephedrine (on average 11 mg to maintain their

**Table 4** Percentage of patients who received oxygen and piritramide (Fisher's exact test).

|  | CLO group | REMI group | *P* |
| --- | --- | --- | --- |
| Number of patients discharged from recovery with oxygen | 3.3% | 24.1% | 0.03[*] |
| Number of patients who received piritramide | 10.0% | 34.5% | 0.03[*] |

Notes.
*Statistically significant.

MAP in the admissible range). In the CLO group, 46.6% of the patients needed ephedrine (on average 5.9 mg) (Table 3).

Time to eye opening, time to extubation, NRS-11 at arrival in the recovery room and time spent in the recovery room did not significantly differ between the two groups. The pain score at discharge from the recovery room was significantly lower in the CLO group, as was the piritramide consumption. The patients from the REMI group had a significantly higher probability to be discharged from the recovery room with supplemental oxygen to maintain their $SpO_2$ within 3% from their preoperative values (Tables 3 and 4).

There was no difference in supplemental oxygen at departure from the recovery room between smokers and non-smokers ($p = 0.67$) or patients having a BMI $\leq 25$ and >25 ($p = 0.12$).

Upon arrival in the recovery room, one patient in the CLO group developed a hypotension (MAP of 60 mmHg) that was successfully treated with a fluid bolus, and one patient in the REMI group developed a sinus tachycardia (HR 120–140/min) that spontaneously ceased after 1 h. No patient suffered from nausea or vomiting, bradycardia, hypertension, and other unwanted side effects.

## DISCUSSION

Many factors are involved in the quality of the surgical field in ESS (e.g., anatomical conditions and severity and inflammatory status of the disease). In this study, these factors have been considered by including the CT-based Mackay-Lund score, a known predictor of bleeding in ESS (*Mortuaire et al., 2008*), and preoperative oral corticoid treatment in the factors used for the minimization. The anesthetic technique also has an impact on the surgeon's comfort. Patient positioning, local anesthetic infiltration, respiratory parameters, and even the way the ETT is secured have been shown to affect blood loss (*Baker & Baker, 2010*).

Both clonidine and remifentanil have been used and studied in the setting of controlled hypotension (*Wawrzyniak, Kusza & Cywinski, 2014*; *Miłoński et al., 2013*; *Mohseni & Ebneshahidi, 2011*; *Marchal et al., 2001*; *Lee et al., 1999*; *Hackmann et al., 2003*; *Yun, Kim & Kim, 2015*; *Cardesin et al., 2015*). These drugs decrease blood loss by decreasing the blood pressure and, more specifically, the cardiac output. *Kazmaier et al. (2000)* found that remifentanil lowers stroke volume and HR in similar proportions, decreasing cardiac output and MAP. Clonidine also decreases blood pressure by lowering cardiac output. Its action is mainly due to the decrease in HR, leaving the stroke volume and the systemic vascular resistance untouched (*Onesti et al., 1971*). I.V. clonidine can produce a transient

hypertensive effect secondary to systemic vasoconstriction, which was not encountered in our study due to the slow speed of the I.V. injection.

In the present study, we showed that both drugs provided the surgeon similar operating conditions with a slightly lower MAP in the REMI group. Ephedrine requirements were significantly higher and more frequent in this group than in the CLO group. One could argue that the remifentanil initial infusion rate was too high. We do not agree with this argument, as the rate was adjusted by a factor of 2 according to the MAP.

The longer half-life of clonidine and sufentanil in comparison to remifentanil as well as the sedative properties of the alpha-2 agonist did not affect the recovery times. Time to eyes opening, time to extubation, and length of stay in the recovery room were not significantly different between both groups. Time to eyes opening and extubation times might seem long, but all anesthetic drug administration was stopped when the surgeon began packing the nose at the very end of the procedure. The study protocol was strict regarding when and how drug administration could be changed. There were no fixed rules for the extubation criteria, which left a certain degree of freedom to the anesthetist in charge.

Patients who received clonidine were less prone to require supplemental oxygen to maintain their $SpO_2$ within 3% of their preoperative values. Clonidine is a drug that has shown a positive effect on the overall saturation on patients suffering from obstructive sleep apnea syndrome (OSAS). This effect was first thought to be the result of the drug's ability to decrease or suppress time spent in REM sleep (*Issa, 1992*). Clonidine has since been studied as a premedication in patients with OSAS and has shown a higher minimal SpO2 on the day of surgery in patients who received clonidine versus placebo (*Pawlik et al., 2005*). The authors explain this difference by the lower amount of propofol and piritramide received by patients with clonidine premedication. In our study, no such difference was observed. The piritramide consumption is statistically different between the two groups but clinically not relevant, and remifentanil did not affect ventilatory mechanics after more than an hour spent in the recovery room. Clonidine is known to decrease the ventilatory response to $CO_2$, which is seen as a central depressant effect by *Ooi, Pattison & Feldman (1991)*, but is considered a protective mechanism against central hypocapnic apnea during non-REM sleep by *Sankri-Tarbichi, Grullon & Badr (2013)*. It remains unclear how clonidine could have a positive effect on awake patients' $SpO_2$ following ESS. We recommend further studies to address this issue. In this study, we did not find a relationship between supplemental oxygen and smoking status or the patient's BMI.

NRS-11 were similar in both groups upon arrival in the recovery room, but were slightly higher in the REMI group upon discharge. Although this difference is statistically significant, it is probably not clinically relevant. Nevertheless, it should be noted that more patients in the REMI group received piritramide at a higher dose than in the CLO group. These results concur with previous results of studies showing that clonidine potentiates opioid drugs or has intrinsic analgesic properties (*Blaudszun et al., 2012*; *Wallace, 2006*). These results could also be explained by the longer acting sufentanil used in the CLO group.

In contrast to *Cardesin et al. (2015)* we did not find an advantage of clonidine over remifentanil for ESS in regard to the surgical field quality. Their anesthesia protocol was slightly different, with lower doses of hypotensive agent (clonidine 2–3 μg/kg vs.

1–1.5 µg/kg and remifentanil 0.34 µg/kg/min vs. 0.1–0.2 µg/kg/min). Cardesin et al. used fentanyl in both group, but we used only sufentanil in the CLO group, which is more rational from a pharmacological point of view and closer to our daily practice. The scores used were different, although loosely comparable as they are both 6-point scales. They were also assessed at different instants. The Boezaart scale is probably less subjective than the Fromme score we used as it is mainly based on the suction usage. Patients and surgical treatments were also different as shown by the Mackay-Lund scores and the mean duration of the surgery. All these elements make it difficult to compare both studies.

## CONCLUSIONS

Although both anesthesia regimens provided similar operating conditions, patients who received clonidine and sufentanil had significantly higher MAP, required less vasopressor, received less piritramide, and had lower NRS-11 scores. Fewer patients in this group needed oxygen to maintain their $SpO_2$ to the preoperative level. The clonidine–sufentanil association has slight advantages, including ease of use, over remifentanil to achieve satisfying operating conditions for patients undergoing ESS for chronic rhinosinusitis.

### Funding
The authors received no funding for this work.

### Competing Interests
The authors declare there are no competing interests.

### Author Contributions
- Laurent Bairy conceived and designed the experiments, performed the experiments, analyzed the data, wrote the paper, prepared figures and/or tables, reviewed drafts of the paper.
- Marie Vanderstichelen conceived and designed the experiments, performed the experiments, analyzed the data, wrote the paper, reviewed drafts of the paper.
- Jacques Jamart conceived and designed the experiments, analyzed the data, contributed reagents/materials/analysis tools, reviewed drafts of the paper.
- Edith Collard conceived and designed the experiments, performed the experiments, wrote the paper, reviewed drafts of the paper.

### Human Ethics
The following information was supplied relating to ethical approvals (i.e., approving body and any reference numbers):

The CHU Mont-Godinne local ethics committee approved the study on (Internal id: 101/2013, Belgian id:B039201318956).

## Clinical Trial Ethics

The following information was supplied relating to ethical approvals (i.e., approving body and any reference numbers):

The CHU Mont-Godinne local ethics committee approved the study (Internal id: 101/2013, Belgian id:B039201318956).

## Data Availability

The raw data has been supplied as a Supplementary File.

## Clinical Trial Registration

The following information was supplied regarding Clinical Trial registration:

ACTRN12614000935639.

## Supplemental Information

Supplemental information for this article can be found online at http://dx.doi.org/10.7717/peerj.3370#supplemental-information.

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
