# Peer review of "Clonidine or remifentanil for adequate surgical conditions in patients undergoing endoscopic sinus surgery: a randomized study"

_PeerJ, doi:10.7717/peerj.3370_

## Round 0.1 · original submission · Major Revisions

Please revise your work according to the reviewer comments by considering all suggestions. In particular the comments of Reviewer 1 must be appropriately addressed.

·

Basic reporting

Formal remark: Abstracts should generally not contain abbreviations. Terms must be spelled in full at least once before using abbreviations. Please have in mind the reader is not always familiar with the topic; the less abbreviations the better.

Introduction: This part should offer a conclusive background for the reader, a requirement that is not adequately met. The reader learns about the special problems of and requirements for sinus surgery, but information about the usual techniques of controlled hypotension is lacking, though there is a large body of data about various effective drugs for this purpose. The authors offer no comprehensible rationale why was the trial performed, and what scientific/clinical benefit was expected.
In addition the introduction lacks exact determination of the objectives of the study, such as 1., 2. etc.

Experimental design

The prospective randomised trial compared sufentanil/clonidine versus Remifentanil anesthesia, both with sevoflurane for sedation in otherwise comparable groups undergoing sinus surgery.
Please explain why clonidine was investigated.
Please explain 'time to eye opening' as well as 'extubation time'. If the respective time interval starts at the end of surgery (last stitch, dressing done) it is quite long in both groups. Please explain.
Methods: The sample size calculation is based on a study which is not further explained.
Except this the methods are adequately explained.

Validity of the findings

Overall both anaesthetic procedures worked well; no relevant complications occurred, no patient was suffering from postoperative nausea and vomiting. Central findings of the study are that patients with remifentanil compared to those with clonidine/sufentanil had a shorter time to eye opening, but similar extubation time; their vasopressor requirement was higher and so was their pain score after surgery, and their requirement of postoperative piritramide. Though patients mean O2Sat in the recovery room was similar between the two groups, at discharge patients with remifentanil required more frequently supplemental oxygen. Finally using remifentanil instead of clonidine/sufentanil is more expensive.
The design of the study leads to results with adequate validity.
Discussion: The authors repeat their findings without discussing crucial issues, for example oxygen demand of patients with remifentanil, an ultra-short acting opiate. Instead they waste 5 references to discuss the oxygen requirement of smokers, which is not relevant in their study.

Additional comments

It is a technically well conducted study, but unfortunately lacking both a comprehensible rationale and an adequate discussion of the findings. I generally sympathize with scientific groups, trying hard to make a nice study, and often have to fight for publication in a good journal, but following a Journal's rules and regulations alone doesn't do the trick.
Study design and research question need some clarification. The authors investigate the effect of a very old, very well known drug, clonidine. Controlled hypotension is applied mainly with inhalation anesthetics or short acting highly effective drugs, such as Nitroglycerin, sodium nitroprusside, adenosine, and Esmolol (1). Clonidine seems to be effective mainly when given before surgery for premedication (2); however, compared to others it has a quite long half-life-time and a slow clearance (s. below). It is definitely not the number one drug of choice for controlled hypotension. In addition Remifentanil is first and foremost a strong analgesic and not an antihypertensive agent. Its cardiovascular side effects can be used and are appreciated in certain situations; however its dosage is determined by the analgesic requirement of the individual patient.
The differences between the groups regarding quality and duration of postoperative analgesia are not surprising, considering the half-life-time of Remi, Sufenta and Clonidine, which is 9, 150, and 420-660 minutes respectively. Moreover, it is well known that the analgesic effect of clonidine lasts 6 - 12 hours (3).
From my point of view, there has to be a comprehensible explanation why the study was done at all (Introduction), and a comprehensive discussion about some of the findings, such as the higher oxygen demand in Remi patients when discharged from PACU, something that is not confirmed within the literature. Why was the extubation time similar, whereas eye opening In Remi patients was about 3 minutes earlier?
Finally elaborate on the financial aspect. Considering costs of machines, maintenance of facilities and most of all manpower, the costs of anesthetic drugs are neglectable.


Drug, Onset of action (min), Half Life (min), Clearance (L/min)
Nitroglycerine 3 - 5, 3, 25
Sodium nitroprusside 1 - 2, 2, 9 - 55
Adenosine 0.1 – 0.35, 0.15, high
Esmolol 2 - 10, 9, 18.5
CLONIDINE i.v. 10, 420 - 660, 0.2

References
1. Degoute CS. Controlled hypotension: a guide to drug choice. Drugs 2007; 67: 1053-76.
2. Wawrzyniak K, Burduk PK, Cywinski JB, Kusza K, Kazmierczak W. Improved quality of surgical field during endoscopic sinus surgery after clonidine premedication--a pilot study. Int.Forum Allergy Rhinol. 2014; 4: 542-7.
3. Bharti N, Dontukurthy S, Bala I, Singh G. Postoperative analgesic effect of intravenous (i.v.) clonidine compared with clonidine administration in wound infiltration for open cholecystectomy. Br.J.Anaesth. 2013; 111: 656-61.

Reviewer 2 ·

Basic reporting

More detailed titles of tables should be provided

Line 78, Fromme score not Fromme score score.

Experimental design

An original research question is clearly defined. The experimental design is appropriate to the question and the methods are well described. Importantly, data are well described for the reader to understand. Ethics approval for the research was obtained from the author's institution. The "Ethics statement" is better to be placed at the end of article.

Validity of the findings

Authors should use median and interquartile range instead of mean ± SD in all tables.

Authors should explain * symbols in Table 2-5.

Authors should write the full name of statistical methods on the titles of Tables 3-4.

The authors use descriptive statistics to present their data. But they should use median and interquartile range instead of mean ± SD because nonparametric test is used in the analysis. And authors should explain why they used nonparametric test? And also, authors should write the full name of statistical methods on the titles of Tables 3-4. Readers can mistake the title "Wilcoxon test".

Reviewer 3 ·

Basic reporting

The structure of the article should conform to an acceptable format of ‘standard sections’ (see our Instructions for Authors for our suggested format). Significant departures in structure should be made only if they significantly improve clarity or conform to a discipline-specific custom

Experimental design

Methods should be described with sufficient information to be reproducible by another investigator.

Validity of the findings

The data on which the conclusions are based must be provided or made available in an acceptable discipline-specific repository. The data should be robust, statistically sound, and controlled

Additional comments

No. Thanks.

---

## Round 0.2 · Minor Revisions

Dear authors, the current version of the ms has been adequately revised according to the reviewer comments! The ms will need some language improvement before it can be published - please revise it for language and resubmit

·

Basic reporting

This is a re-review after the authors did a thorough revision of the manuscript.

I re-checked the attached article-with-changes-tracking.

Experimental design

This is a re-review

Validity of the findings

This is a re-review

Additional comments

Generally the requirements have been complied, and the paper can be published without further review.

However, before finally submitting, the authors have to check the manuscript for orthographic errors(!), as there are some, already indicated by the computer's language control.

---

## Round 0.3 · accepted · Accept

The manuscript is now ready to be published in PeerJ.